# EMERGENT COMMUNICATION FOR UNDERSTANDING HUMAN LANGUAGE EVOLUTION: WHAT'S MISSING?

**Lukas Galke**
Max Planck Institute for Psycholinguistics
Nijmegen, Netherlands
lukas.galke@mpi.nl

**Yoav Ram**
School of Zoology
Faculty of Life Sciences
Sagol School of Neuroscience
Tel Aviv University, Israel
yoav@yoavram.com

**Limor Raviv**
Max Planck Institute for Psycholinguistics
Nijmegen, Netherlands
Centre for Social, Cognitive and Affective Neuroscience
University of Glasgow, Scotland
limor.raviv@mpi.nl

## ABSTRACT

Emergent communication protocols among humans and artificial neural network agents do not yet share the same properties and show some critical mismatches in results. We describe three important phenomena with respect to the emergence and benefits of compositionality: ease-of-learning, generalization, and group size effects (i.e., larger groups create more systematic languages). The latter two are not fully replicated with neural agents, which hinders the use of neural emergent communication for language evolution research. We argue that one possible reason for these mismatches is that key cognitive and communicative constraints of humans are not yet integrated. Specifically, in humans, memory constraints and the alternation between the roles of speaker and listener underlie the emergence of linguistic structure, yet these constraints are typically absent in neural simulations. We suggest that introducing such communicative and cognitive constraints would promote more linguistically plausible behaviors with neural agents.

## 1 INTRODUCTION

Emergent communication has been widely studied in deep learning (Lazaridou & Baroni, 2020), and in language evolution (Selten & Warglien, 2007; Winters et al., 2015; Raviv et al., 2019). Both fields share communication games as a common experimental framework: a speaker describes an input, e. g., an object or a scene, and transmits a message to a listener, which has to guess or reconstruct the speaker's input. However, the languages of artificial neural network agents (neural agents) do not always exhibit the same linguistic properties as human languages. This presents a problem for using emergent communication as a model for language evolution of humans (or animals (Prat, 2019)).

Here, we emphasize three important phenomena in human language evolution (described in detail in Section 2) that relate to the emergence of compositional structure — all of which have been discussed theoretically and confirmed experimentally with humans. First, compositional languages are easier to learn (Kirby et al., 2014; Carr et al., 2017; Raviv et al., 2021). Second, more compositionality allows for better generalization and facilitates convergence between strangers (Wray & Grace, 2007; Raviv et al., 2021). Third, larger populations generally develop more structured languages (Lupyan & Dale, 2010; Raviv et al., 2019).

However, in emergent communication between neural agents, two of the three phenomena are not yet replicated (see Section 3). Although the ease-of-learning effect for compositional structure has been confirmed in multiple experiments (Li & Bowling, 2019; Guo et al., 2019; Chaabouni et al., 2020), recent work has shown that compositional structure is not necessary for generalization (Lazaridou

et al., 2018; Chaabouni et al., 2020). Regarding the effect of group size, so far this could only be confirmed with continuous communication channels (Tieleman et al., 2019). With discrete communication, an increase in group size does not lead to the emergence of more compositional languages (Chaabouni et al., 2022). Only recently, Rita et al. (2022) have shown that the group size effect can be partially recovered by explicitly modeling population heterogeneity.

We propose two potential explanations for the striking mismatches between humans and neural agents. We argue that emergent communication simulations with neural networks typically lack two key communicative and cognitive factors that drive the emergence of compositionality in humans, and whose omission essentially eliminates the benefits of compositionality: memory constraints, and alternating roles. These are described in detail in Section 4, along with suggestions on how to tackle them, namely, limiting model capacity and sharing parameters within single agents.

## 2    IMPORTANT PROPERTIES OF HUMAN LANGUAGES

Two of the most fundamental properties of human languages are 1. a consistent form-to-meaning mapping and 2. a compositional structure (Hockett, 1960). Previous work suggests that compositional structure evolves as the trade-off between a compressability pressure (i.e., languages should be as simple and learnable as possible) and an expressivity pressure (i.e., languages should be able to successfully disambiguate between a variety of meanings) (Kirby et al., 2015). Without a compressibility pressure, languages would become completely holistic (that is, with a unique symbol for each meaning) - which is highly expressive but poorly compressed. Without an expressivity pressure, languages would become degenerate (that is, comprised of a single symbol) - which is highly compressed but not expressive. Yet, with both pressures present, as in the case of natural languages, structured languages with compositionality would emerge — as these strike a balance between compressibility and expressivity. Notably, three phenomena related to compositionality have been demonstrated and discussed with humans:

**Compositional languages are easier to learn.**    The driving force behind a compressibility pressure is that languages must be transmitted, learned, and used by multiple individuals, often from limited input and with limited exposure time (Smith et al., 2003). Numerous iterated learning studies have shown that artificial languages become easier to learn over time (Kirby et al., 2008; Winters et al., 2015; Beckner et al., 2017) — a finding that is attributed to a simultaneous increase in compositionality. Indeed, artificial language learning experiments directly confirm that more compositional languages are learned better and faster by adults (Raviv et al., 2021).

**Compositionality aids generalization.**    When a language exhibits a compositional structure, it is easier to generalize (describe new meanings) in a way that is transparent and understandable to other speakers (Kirby, 2002; Zuidema, 2002). For example, the need to communicate over a growing number of different items in an open-ended meaning space promotes the emergence of more compositional systems Carr et al. (2017). Recently, Raviv et al. (2021) showed that compositional structure predicts generalization performance, with compositional languages that allow better generalizations that are also shared across different individuals .The rationale is that humans cannot remember a holistic language (compressability pressure), and when they need to invent descriptions for new meanings, compositionality enables systematic and transparent composition of new label-meaning pairs from existing part-labels.

**Larger groups create more compositional languages.**    Socio-demographic factors such as population size have long been assumed to be important determinants of language evolution (Wray & Grace, 2007; Nettle, 2012; Lupyan & Dale, 2010). Specifically, global cross-linguistic studies found that bigger communities tend to have languages with more systematic and transparent structures (Lupyan & Dale, 2010). This result has also been experimentally confirmed, where larger groups of interacting participants created more compositional languages (Raviv et al., 2019). These findings are attributed to compressibility pressures arising during communication: remembering partner-specific variants becomes increasingly more challenging as group size increases, and so memory constraints force larger groups to converge on more transparent languages.

## 3 REPLICATION ATTEMPTS WITH NEURAL AGENTS

Computational modeling has long been used to study language evolution (Kirby, 2002; Smith et al., 2003; Kirby et al., 2004; Smith & Kirby, 2008; Gong et al., 2008; Dale & Lupyan, 2012; Perfors & Navarro, 2014; Kirby et al., 2015; Steels, 2016). More recently, emergent communication with neural networks and reinforcement learning techniques was introduced (Lazaridou et al., 2017; Havrylov & Titov, 2017; Kottur et al., 2017). Table 1 offers a (non-exhaustive) summary of recent emergent communication experiments with neural agents. Both symbolic and visual inputs have been used (Lazaridou et al., 2018), and the channel bandwidth (alphabet size to the power of message length) has been increasing with time. Most experiments use long short-term memory (LSTM) (Hochreiter & Schmidhuber, 1997) as the agents' architecture, and training is most often carried out by the REINFORCE algorithm (Williams, 1992). Crucially, most experiments have been limited to a pair of agents, although Li & Bowling (2019) experimented with a single speaker and multiple listeners. Only recently, Chaabouni et al. (2022) and Rita et al. (2022) scaled up the group size. In all experiments except Graesser et al. (2019), agents do not alternate between the roles of speaker and listener and instead employ distinct models for speaker and listener. For comparison, experiments with humans by Raviv et al. (2019) have had $4 \times 16$ different visual meanings, a discrete communication channel with bandwidth $16^{16}$, and group sizes of 4-8 participants, who alternated between being speakers and listeners.

Table 1: Neural emergent communication experiments. Input type and size of meaning space, channel bandwidth (alphabet size to the power of message length), task objective (reconstruction and discrimination, the latter with number of distractors), group size, role alternation between speaker and listener (RA), and presence of iterated learning (IL). Placeholders like N, K, or comma-separated lists mean that settings were varied.

| Experiment | Inputs | Channel | Objective | Groups | RA | IL |
|---|---|---|---|---|---|---|
| Havrylov & Titov (2017) | visual (MSCOCO) | $128^N$ | Discr. (128) | 2 | No | No |
| Kottur et al. (2017) | symbolic ($4^3$) | $N^1$ | Reconstr. | 2 | No | No |
| Lazaridou et al. (2017) | pretrained visual | $10^1/100^1$ | Discr. | 2 | No | No |
| Lazaridou et al. (2018) | symbolic (463) | $10^2/17^5/40^{10}$ | Discr. (5) | 2 | No | No |
| Lazaridou et al. (2018) | visual ($124 \times 124$) | N/A | Discr. (2,20) | 2 | No | No |
| Tieleman et al. (2019) | visual | *continuous* | Reconstr. | 1, 2, 4, 8, 16, 32 | No | No |
| Chaabouni et al. (2019) | symbolic(K) | $40^{30}$ | Reconstr. | 50, 100 | No | No |
| Graesser et al. (2019) | visual | $2^8$ | Discr. | $N$ | Yes | No |
| Rita et al. (2020) | symbolic (1000) | $40^{30}$ | Reconstr. | 2 | No | No |
| Li & Bowling (2019) | symbolic ($8 \times 4$) | $8^2$ | Discr. (5) | $1 : 1, 2, 10$ | No | Yes |
| Kharitonov et al. (2020) | symbolic ($2^8$) | 1 | Partial Reconstr. | 2 | No | No |
| Kharitonov et al. (2020) | visual ($10^2$) | $2^{10}$ | Discr. (100) | 2 | No | No |
| Chaabouni et al. (2020) | symbolic ($2 \times 100$) | $100^3$ | Reconstr. | 2 | No | Yes |
| Chaabouni et al. (2022) | visual (ImageNet, CelebA) | $20^{10}$ | Discr. (20–1024) | 2, 20, 100 | No | Yes |
| Rita et al. (2022) | symbolic ($4 \times 4$) | $20^{10}$ | Reconstr. | 2,10 | No | No |

With respect to the three linguistic phenomena described above for human participants, neural agents show a mixed pattern of results: First, ease-of-learning of compositional languages has been confirmed in emergent communication with neural agents (Guo et al., 2019; Chaabouni et al., 2020; 2022). For example, more compositionality emerges when agents are being constantly reset and need to learn the language over and over again (similar to the iterated learning paradigm) (Li & Bowling, 2019). Similarly, Guo et al. (2019) found that compositional languages emerge in iterated learning experiments with neural agents because they are easier to learn.

Second, in contrast to humans, neural agents can generalize well even without compositional communication protocols. Chaabouni et al. (2020) have found that compositionality is *not* necessary for generalization in neural agents (in line with earlier findings from Lazaridou et al. (2018)). Although they argue that structure (however it emerges) prevails throughout evolution *because* of its implied learnability advantage (in line with Kirby et al. (2015)), the finding that compositionality aids generalization has nevertheless not been replicated with neural agents yet.

Third, in populations of autoencoders, where the encoder and decoders were decoupled and exchanged throughout training, larger communities produced representations with less idiosyncrasies (Tieleman et al., 2019). However, these experiments used a continuous, rather than discrete, channel, which has only recently been analyzed with an increasing population size (Chaabouni et al.,

2022; Rita et al., 2022). Although Chaabouni et al. (2022) argue that it is necessary to scale up emergent communication experiments to better align neural emergent communication with human language evolution, they have not found a systematic advantage of population size in generalization and ease-of-learning (in contrast with (Tieleman et al., 2019)). Similarly, Rita et al. (2022) found that language properties are not enhanced by population size alone. However, when adding heterogeneity through different learning rates, an increase in population size led to an increase in structure.

## 4    POTENTIAL REASONS FOR THE MISMATCH IN RESULTS

Why does the population size not affect emergent neural communication? And why do neural agents not need compositionality to generalize? Our key argument is that crucial communicative and cognitive factors in humans have not yet been appropriately introduced to neural agents in emergent communication experiments. We argue that omitting these factors effectively removes the compressibility pressure that underlies the emergence and benefits of compositionality. In the following, we highlight two crucial factors: memory constraints and speaker/listener role alternation.

**Memory Constraints**    Human memory limitation is one of the most important constraints of language learning, and underlies the compressibility pressure in language use and transmission. Indeed, sequence memory constraints induce structure emergence in iterated learning Cornish et al. (2017), and underlie group size effects in real-world settings (Meir et al., 2012; Wray & Grace, 2007) and in communication games with humans (Raviv et al., 2019), where more compositionality is promoted because individuals cannot memorize partner-specific variations as the group size increases. In contrast, a key ingredient for the success of neural networks is their overparameterization (Nakkiran et al., 2021), which means that their capacity is in fact sufficient to completely "memorize" sender-specific idiosyncratic languages. We propose that this overparameterization significantly reduces compressibility pressure, effectively eliminating the potential benefits of compositional structure for learning and generalization.

Therefore, we suggest that in communication games, the model capacity, i. e., how much information the model can store (well quantifiable, e. g., see MacKay et al. (2003)), should be considered in relation to the number of different meanings and the space of all possible messages, i. e. alphabet size to the power of (maximum) message length $|\mathcal{A}|^L$. We hypothesize that for compositionality to emerge, the model capacity needs to be less than required to memorize a separate message for each meaning from every agent, but also less than the theoretical channel bandwidth, such that it becomes necessary to reuse substructures within the messages. Consistent with our position, Resnick et al. (2020) and Gupta et al. (2020) verified that learning an underlying compositional structure requires less capacity than memorizing a dataset. Similarly to us, the authors of both works assume a range in model capacity that facilitates compositionality, but so far only determine a lower bound, while we argue here about the upper bound(s).

**Role Alternation**    In current neural emergent communication experiments, one agent generates the message, and the other only processes it. This in sharp contrast to human communication, where speakers can reproduce any linguistic message that they can understand (Hockett, 1960). Indeed, dyadic and group communication experiments with humans typically have people alternating between the roles of speaker and listener throughout the experiment (Kirby et al., 2015; Raviv et al., 2019; Motamedi et al., 2021). This is only rarely reflected in work with neural agents (Table 1).

A straight-forward implementation of role alternation is to have shared parameters within the (generative) speaker role and the (discriminative) listener role of the same neural agent. This would make the architecture of neural agents more similar to the human brain, where shared neural mechanisms support both the production and the comprehension of natural speech (Silbert et al., 2014). One way to do this would be to tie the output layer's weights to input embedding, a well known concept in language modeling (Mikolov et al., 2013; Raffel et al., 2020). Some experiments already implement role alternation, e. g., in multi-agent communication with given language descriptions (Graesser et al., 2019), or in language acquisition from image captions where agents simultaneously learn by cognition and production (Nikolaus & Fourtassi, 2021). We suggest role alternation should also implemented in emergent communication experiments to ensure more linguistically plausible dynamics.

## 5 CONCLUSION

We have outlined important discrepancies in the results between emergent communication with human versus neural agents and suggested that these can be explained by the absence of key cognitive and communicative factors that drive human language evolution: memory constraints and speaker-listener role alternation. We suggest that including these factors in future work would mimic the compressability pressure and compositionality benefits observed with human agents, and consequentially would make emergent neural communication protocols more linguistically plausible. Notably, additional psycho- and socio-linguistic factors may affect language evolution, and might also play a role in explaining the discrepancy in results.

## ACKNOWLEDGEMENTS

We thank Mitja Nikolaus for valuable discussions on role alternation and parameter sharing. This research is partly funded by Minerva Center for Lab Evolution; John Templeton Foundation grant.

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
