# OpenReview forum: "Emergent Communication for Understanding Human Language Evolution: What's Missing?"
_ICLR.cc/2022/Workshop/EmeCom — EmeCom Workshop at ICLR 2022_

### Official Review · Reviewer_gmT5 · 2022-03-21
**A clear positional paper that argues the importance of incorporating memory constraints and alternating roles in emergent communication simulation to understand human language evolution**

**Rating:** Accept
**Confidence:** 4

**Review:**

This paper gives a nice overview of the emergent communication field comparing different existing works. It suggests that the mismatch observed between neural emergent communication and language evolution research could be explained by the lack of fundamental cognitive modeling (1) memory constraints and (2) alternating roles. Authors justify their hypothesis from experiments in the cognitive science community. Hence, I believe that this paper can create interesting discussions for the emergent communication field.

---

### Official Review · Reviewer_HqZ6 · 2022-03-22
**Well-written discussion on the need of incorporating realistic constraints in EC simulations in order to reduce the gap between neural EC results and sociolinguistics evidences**

**Rating:** Accept
**Confidence:** 5

**Review:**

**Summary**

This paper notes that results obtained in neural emergent communication often do not fit broader linguistic literature evidences. In particular, it notices that (1) the impact of group size on compositionality and (2) the gain of generalization due to compositionality are not fully replicated with artificial agents. After comparing the results obtained in neural EC literature with cognitive science and sociolinguistics evidences, they propose some updates of current neural EC models : the addition of memory constraints and the alternation between speaker and listener role in simulations.


**Comments**

This paper is a nice comparison between results seen in cognitive science/sociolinguistics and results obtained with neural EC simulations. Rightly, the paper focuses on specific questions (memory constraints, role alternation). It enables the authors to go into details on:
- what should be expected from cogsci/sociolinguistic results
- what has been reproduced/not reproduced with neural EC simulations
- first ideas on what should be updated to improve current models

The paper is really easy to follow and I see it as a very nice starting point for discussions on the interest of using artificial agents as a simulation tool for language evolution.

For all these reasons, I think this paper should be accepted as it will initiate relevant debates for the (neural) EC community.

---

### Decision · Program_Chairs · 2022-03-25

**Decision:**

Accept

**Comment:**

An excellent positional paper arguing for adding constraints to close the gap between real and simulated communication outcomes. Looking forward to the discussions!